# Heavy Ball Neural Ordinary Differential Equations

**Hedi Xia**[*]
Department of Mathematics
University of California, Los Angeles

**Vai Suliafu** [*]
Scientific Computing and Imaging (SCI) Institute
University of Utah, Salt Lake City, UT, USA

**Hangjie Ji, Tan M. Nguyen, Andrea L. Bertozzi, and Stanley J. Osher**
Department of Mathematics
University of California, Los Angeles

**Bao Wang** [†]
Department of Mathematics
Scientific Computing and Imaging (SCI) Institute
University of Utah, Salt Lake City, UT, USA

## Abstract

We propose heavy ball neural ordinary differential equations (HBNODEs), leveraging the continuous limit of the classical momentum accelerated gradient descent, to improve neural ODEs (NODEs) training and inference. HBNODEs have two properties that imply practical advantages over NODEs: (i) The adjoint state of an HBNODE also satisfies an HBNODE, accelerating both forward and backward ODE solvers, thus significantly reducing the number of function evaluations (NFEs) and improving the utility of the trained models. (ii) The spectrum of HBNODEs is well structured, enabling effective learning of long-term dependencies from complex sequential data. We verify the advantages of HBNODEs over NODEs on benchmark tasks, including image classification, learning complex dynamics, and sequential modeling. Our method requires remarkably fewer forward and backward NFEs, is more accurate, and learns long-term dependencies more effectively than the other ODE-based neural network models. Code is available at `https://github.com/hedixia/HeavyBallNODE`.

## 1   Introduction

Neural ordinary differential equations (NODEs) are a family of continuous-depth machine learning (ML) models whose forward and backward propagations rely on solving an ODE and its adjoint equation [4]. NODEs model the dynamics of hidden features $\boldsymbol{h}(t) \in \mathbb{R}^N$ using an ODE, which is parametrized by a neural network $f(\boldsymbol{h}(t), t, \theta) \in \mathbb{R}^N$ with learnable parameters $\theta$, i.e.,

$$\frac{d\boldsymbol{h}(t)}{dt} = f(\boldsymbol{h}(t), t, \theta). \tag{1}$$

Starting from the input $\boldsymbol{h}(t_0)$, NODEs obtain the output $\boldsymbol{h}(T)$ by solving (1) for $t_0 \leq t \leq T$ with the initial value $\boldsymbol{h}(t_0)$, using a black-box numerical ODE solver. The number of function evaluations (NFEs) that the black-box ODE solver requires in a single forward pass is an analogue for the continuous-depth models [4] to the depth of networks in ResNets [16]. The loss between NODE prediction $\boldsymbol{h}(T)$ and the ground truth is denoted by $\mathcal{L}(\boldsymbol{h}(T))$; we update parameters $\theta$ using the following gradient

[*]Co-first author

[†]Please correspond to: wangbaonj@gmail.com

$$\frac{d\mathcal{L}(\boldsymbol{h}(T))}{d\theta} = \int_{t_0}^{T} \boldsymbol{a}(t) \frac{\partial f(\boldsymbol{h}(t), t, \theta)}{\partial \theta} dt, \quad (2)$$

where $\boldsymbol{a}(t) := \partial\mathcal{L}/\partial\boldsymbol{h}(t)$ is the adjoint state, which satisfies the following adjoint equation

$$\frac{d\boldsymbol{a}(t)}{dt} = -\boldsymbol{a}(t) \frac{\partial f(\boldsymbol{h}(t), t, \theta)}{\partial \boldsymbol{h}}. \quad (3)$$

NODEs are flexible in learning from irregularly-sampled sequential data and particularly suitable for learning complex dynamical systems [4, 42, 56, 35, 9, 24], which can be trained by efficient algorithms [40, 7, 58]. NODE-based continuous generative models have computational advantages over the classical normalizing flows [4, 15, 55, 12]. NODEs have also been generalized to neural stochastic differential equations, stochastic processes, and graph

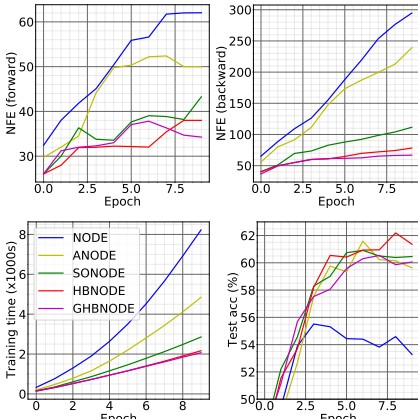

Figure 1: Contrasting NODE, ANODE, SONODE, HBNODE, and GHBNODE for CIFAR10 classification in NFEs, training time, and test accuracy. (Tolerance: $10^{-5}$, see Sec. 5.2 for experimental details.)

NODEs [21, 28, 38, 49, 20, 34]. The drawback of NODEs is also prominent. In many ML tasks, NODEs require very high NFEs in both training and inference, especially in high accuracy settings where a lower tolerance is needed. The NFEs increase rapidly with training; high NFEs reduce computational speed and accuracy of NODEs and can lead to blow-ups in the worst-case scenario [15, 10, 29, 35]. As an illustration, we train NODEs for CIFAR10 classification using the same model and experimental settings as in [10], except using a tolerance of $10^{-5}$; Fig. 1 shows both forward and backward NFEs and the training time of different ODE-based models; we see that NFEs and computational times increase very rapidly for NODE, ANODE [10], and SONODE [35]. More results on the large NFE and degrading utility issues for different benchmark experiments are available in Sec. 5. Another issue is that NODEs often fail to effectively learn long-term dependencies in sequential data [26], as discussed in Sec. 4.

## 1.1 Contribution

We propose heavy ball neural ODEs (HBNODEs), leveraging the continuous limit of the classical momentum accelerated gradient descent, to improve NODE training and inference. At the core of HBNODE is replacing the first-order ODE (1) with a heavy ball ODE (HBODE), i.e., a second-order ODE with an appropriate damping term. HBNODEs have two theoretical properties that imply practical advantages over NODEs:

- The adjoint equation used for training a HBNODE is also a HBNODE (see Prop. 1 and Prop. 2), accelerating both forward and backward propagation, thus significantly reducing both forward and backward NFEs. The reduction in NFE using HBNODE over existing benchmark ODE-based models becomes more aggressive as the error tolerance of the ODE solvers decreases.

- The spectrum of the HBODE is well-structured (see Prop. 4), alleviating the vanishing gradient issue in back-propagation and enabling the model to effectively learn long-term dependencies from sequential data.

To mitigate the potential blow-up problem in training HBNODEs, we further propose generalized HBNODEs (GHBNODEs) by integrating skip connections [17] and gating mechanisms [19] into the HBNODE. See Sec. 3 for details.

## 1.2 Organization

We organize the paper as follows: In Secs. 2 and 3, we present our motivation, algorithm, and analysis of HBNODEs and GHBNODEs, respectively. We analyze the spectrum structure of the adjoint equation of HBNODEs/GHBNODEs in Sec. 4, which indicates that HBNODEs/GHBNODEs can learn long-term dependency effectively. We test the performance of HBNODEs and GHBNODEs on benchmark point cloud separation, image classification, learning dynamics, and sequential modeling in Sec. 5. We discuss more related work in Sec. 6, followed by concluding remarks. Technical proofs and more experimental details are provided in the appendix.

## 2 Heavy Ball Neural Ordinary Differential Equations

### 2.1 Heavy ball ordinary differential equation

Classical momentum method, a.k.a., the heavy ball method, has achieved remarkable success in accelerating gradient descent [39] and has significantly improved the training of deep neural networks [46]. As the continuous limit of the classical momentum method, heavy ball ODE (HBODE) has been studied in various settings and has been used to analyze the acceleration phenomenon of the momentum methods. For the ease of reading and completeness, we derive the HBODE from the classical momentum method. Starting from initial points $x^0$ and $x^1$, gradient descent with classical momentum searches a minimum of the function $F(x)$ through the following iteration

$$x^{k+1} = x^k - s\nabla F(x^k) + \beta(x^k - x^{k-1}), \tag{4}$$

where $s > 0$ is the step size and $0 \leq \beta < 1$ is the momentum hyperparameter. For any fixed step size $s$, let $m^k := (x^{k+1} - x^k)/\sqrt{s}$, and let $\beta := 1 - \gamma\sqrt{s}$, where $\gamma \geq 0$ is another hyperparameter. Then we can rewrite (4) as

$$m^{k+1} = (1 - \gamma\sqrt{s})m^k - \sqrt{s}\nabla F(x^k); \ x^{k+1} = x^k + \sqrt{s}m^{k+1}. \tag{5}$$

Let $s \to 0$ in (5); we obtain the following system of first-order ODEs,

$$\frac{dx(t)}{dt} = m(t); \ \frac{dm(t)}{dt} = -\gamma m(t) - \nabla F(x(t)). \tag{6}$$

This can be further rewritten as a second-order heavy ball ODE (HBODE), which also models a damped oscillator,

$$\frac{d^2 x(t)}{dt^2} + \gamma\frac{dx(t)}{dt} = -\nabla F(x(t)). \tag{7}$$

In Appendix E.6, we compare the dynamics of HBODE (7) and the following ODE limit of the gradient descent (GD)

$$\frac{dx}{dt} = -\nabla F(x). \tag{8}$$

In particular, we solve the ODEs (7) and (8) with $F(x)$ defined as a `Rosenbrock` [41] or `Beale` [14] function. The comparisons show that HBODE can accelerate the dynamics of the ODE for a gradient system, which motivates us to propose HBNODE to accelerate forward propagation of NODE.

### 2.2 Heavy ball neural ordinary differential equations

Similar to NODE, we parameterize $-\nabla F$ in (7) using a neural network $f(h(t), t, \theta)$, resulting in the following HBNODE with initial position $h(t_0)$ and momentum $m(t_0) := dh/dt(t_0)$,

$$\frac{d^2 h(t)}{dt^2} + \gamma\frac{dh(t)}{dt} = f(h(t), t, \theta), \tag{9}$$

where $\gamma \geq 0$ is the damping parameter, which can be set as a tunable or a learnable hyperparmater with positivity constraint. In the trainable case, we use $\gamma = \epsilon \cdot \text{sigmoid}(\omega)$ for a trainable $\omega \in \mathbb{R}$ and a fixed tunable upper bound $\epsilon$ (we set $\epsilon = 1$ below). According to (6), HBNODE (9) is equivalent to

$$\frac{dh(t)}{dt} = m(t); \quad \frac{dm(t)}{dt} = -\gamma m(t) + f(h(t), t, \theta). \tag{10}$$

Equation (9) (or equivalently, the system (10)) defines the forward ODE for the HBNODE, and we can use either the first-order (Prop. 2) or the second-order (Prop. 1) adjoint sensitivity method to update the parameter $\theta$ [35].

**Proposition 1** (Adjoint equation for HBNODE). *The adjoint state $a(t) := \partial\mathcal{L}/\partial h(t)$ for the HBNODE (9) satisfies the following HBODE with the same damping parameter $\gamma$ as that in (9),*

$$\frac{d^2 a(t)}{dt^2} - \gamma\frac{da(t)}{dt} = a(t)\frac{\partial f}{\partial h}(h(t), t, \theta). \tag{11}$$

**Remark 1.** *Note that we solve the adjoint equation (11) from time $t = T$ to $t = t_0$ in the backward propagation. By letting $\tau = T - t$ and $b(\tau) = a(T - \tau)$, we can rewrite (11) as follows,*

$$\frac{d^2 b(\tau)}{d\tau^2} + \gamma\frac{db(\tau)}{d\tau} = b(\tau)\frac{\partial f}{\partial h}(h(T - \tau), T - \tau, \theta). \tag{12}$$

*Therefore, the adjoint of the HBNODE is also a HBNODE and they have the same damping parameter.*

We can also employ (10) and its adjoint for the forward and backward propagations, respectively.

**Proposition 2** (Adjoint equations for the first-order HBNODE system). *The adjoint states* $\boldsymbol{a_h}(t)$ $:= \partial\mathcal{L}/\partial\boldsymbol{h}(t)$ *and* $\boldsymbol{a_m}(t) := \partial\mathcal{L}/\partial\boldsymbol{m}(t)$ *for the first-order HBNODE system* (10) *satisfy*

$$\frac{d\boldsymbol{a_h}(t)}{dt} = -\boldsymbol{a_m}(t)\frac{\partial f}{\partial\boldsymbol{h}}(\boldsymbol{h}(t), t, \theta); \quad \frac{d\boldsymbol{a_m}(t)}{dt} = -\boldsymbol{a_h}(t) + \gamma\boldsymbol{a_m}(t). \tag{13}$$

**Remark 2.** *Let* $\tilde{\boldsymbol{a}}_{\boldsymbol{m}}(t) = d\boldsymbol{a_m}(t)/dt$, *then* $\boldsymbol{a_m}(t)$ *and* $\tilde{\boldsymbol{a}}_{\boldsymbol{m}}(t)$ *satisfies the following first-order heavy ball ODE system*

$$\frac{d\boldsymbol{a_m}(t)}{dt} = \tilde{\boldsymbol{a}}_{\boldsymbol{m}}(t); \quad \frac{d\tilde{\boldsymbol{a}}_{\boldsymbol{m}}(t)}{dt} = \boldsymbol{a_m}(t)\frac{\partial f}{\partial\boldsymbol{h}}(\boldsymbol{h}(t), t, \theta) + \gamma\tilde{\boldsymbol{a}}_{\boldsymbol{m}}(t). \tag{14}$$

*Note that we solve this system backward in time in back-propagation. Moreover, we have* $\boldsymbol{a_h}(t) = \gamma\boldsymbol{a_m}(t) - \tilde{\boldsymbol{a}}_{\boldsymbol{m}}(t)$.

Similar to [35], we use the coupled first-order HBNODE system (10) and its adjoint first-order HBNODE system (13) for practical implementation, since the entangled representation permits faster computation [35] of the gradients of the coupled ODE systems.

## 3 Generalized Heavy Ball Neural Ordinary Differential Equations

In this section, we propose a generalized version of HBNODE (GHBNODE), see (15), to mitigate the potential blow-up issue in training ODE-based models. In our experiments, we observe that $\boldsymbol{h}(t)$ of ANODEs [10], SONODEs [35], and HBNODEs (10) usually grows much faster than that of NODEs. The fast growth of $\boldsymbol{h}(t)$ can lead to finite-time blow up. As an illustration, we compare the performance of NODE, ANODE, SON-ODE, HBNODE, and GHBNODE on the Silverbox task as in [35]. The goal of the task is to learn the voltage of an electronic circuit that resembles a Duffing oscillator, where the input voltage $V_1(t)$ is used to predict the output $V_2(t)$. Similar to the setting in [35], we first augment ANODE by 1 dimension with 0-augmentation and augment SONODE, HBNODE, and GHBNODE

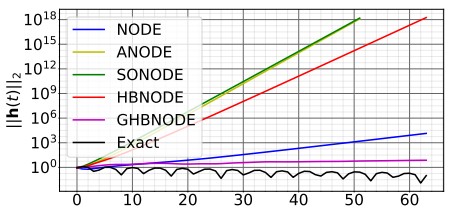

Figure 2: Contrasting $\boldsymbol{h}(t)$ for different models. $\boldsymbol{h}(t)$ in ANODE, SONODE, and HBNODE grows much faster than that in NODE. GHBNODE controls the growth of $\boldsymbol{h}(t)$ effectively when $t$ is large.

with a dense network. We use a simple dense layer to parameterize $f$ for all five models, with an extra input term for $V_1(t)^3$. For both HBNODE and GHBNODE, we set the damping parameter $\gamma$ to be $\mathrm{sigmoid}(-3)$. For GHBNODE (15) below, we set $\sigma(\cdot)$ to be the `hardtanh` function with bound $[-5, 5]$ and $\xi = \ln(2)$. The detailed architecture can be found in Appendix E. As shown in Fig. 2, compared to the vanilla NODE, the $\ell_2$ norm of $\boldsymbol{h}(t)$ grows much faster when a higher order NODE is used, which leads to blow-up during training. Similar issues arise in the time series experiments (see Sec. 5.4), where SONODE blows up during long term integration in time, and HBNODE suffers from the same issue with same initialization.

To alleviate the problem above, we propose the following generalized HBNODE

$$\frac{d\boldsymbol{h}(t)}{dt} = \sigma(\boldsymbol{m}(t)); \quad \frac{d\boldsymbol{m}(t)}{dt} = -\gamma\boldsymbol{m}(t) + f(\boldsymbol{h}(t), t, \theta) - \xi\boldsymbol{h}(t), \tag{15}$$

where $\sigma(\cdot)$ is a nonlinear activation, which is set as $\tanh$ in our experiments. The positive hyperparameters $\gamma, \xi > 0$ are tunable or learnable. In the trainable case, we let $\gamma = \epsilon \cdot \mathrm{sigmoid}(\omega)$ as in HBNODE, and $\xi = \mathrm{softplus}(\chi)$ to ensure that $\gamma, \xi \geq 0$. Here, we integrate two main ideas into the design of GHBNODE: (i) We incorporate the gating mechanism used in LSTM [19] and GRU [6], which can suppress the aggregation of $\boldsymbol{m}(t)$; (ii) Following the idea of skip connection [17], we add the term $\xi\boldsymbol{h}(t)$ into the governing equation of $\boldsymbol{m}(t)$, which benefits training and generalization of GHBNODEs. Fig. 2 shows that GHBNODE can indeed control the growth of $\boldsymbol{h}(t)$ effectively.

**Proposition 3** (Adjoint equations for GHBNODEs). *The adjoint states* $\boldsymbol{a_h}(t) := \partial\mathcal{L}/\partial\boldsymbol{h}(t)$, $\boldsymbol{a_m}(t) := \partial\mathcal{L}/\partial\boldsymbol{m}(t)$ *for the GHBNODE* (15) *satisfy the following first-order ODE system*

$$\frac{\partial\boldsymbol{a_h}(t)}{\partial t} = -\boldsymbol{a_m}(t)\Big(\frac{\partial f}{\partial\boldsymbol{h}}(\boldsymbol{h}(t), t, \theta) - \xi\boldsymbol{I}\Big), \quad \frac{\partial\boldsymbol{a_m}(t)}{\partial t} = -\boldsymbol{a_h}(t)\sigma'(\boldsymbol{m}(t)) + \gamma\boldsymbol{a_m}(t). \tag{16}$$

---

[3]Here, we exclude an $\boldsymbol{h}^3$ term that appeared in the original Duffing oscillator model because including it would result in finite-time explosion.

Though the adjoint state of the GHBNODE (16) does not satisfy the exact heavy ball ODE, based on our empirical study, it also significantly reduces the backward NFEs.

## 4   Learning long-term dependencies – Vanishing gradient

It is known that the vanishing and exploding gradients are two bottlenecks for training recurrent neural networks (RNNs) with long-term dependencies [2, 37] (see Appendix C for a brief review on the exploding and vanishing gradient issues in training RNNs). The exploding gradients issue can be effectively resolved via gradient clipping, training loss regularization, etc [37, 11]. Thus in practice the vanishing gradient is the major issue for learning long-term dependencies [37]. As the continuous analogue of RNN, NODEs as well as their hybrid ODE-RNN models, may also suffer from vanishing in the adjoint state $\boldsymbol{a}(t) := \partial \mathcal{L} / \partial \boldsymbol{h}(t)$ [26]. When the vanishing gradient issue happens, $\boldsymbol{a}(t)$ goes to $\boldsymbol{0}$ quickly as $T - t$ increases, then $d\mathcal{L}/d\theta$ in (2) will be independent of these $\boldsymbol{a}(t)$. We have the following expressions for the adjoint states of the NODE and HBNODE (see Appendix C for details):

- For NODE, we have

$$
\frac{\partial \mathcal{L}}{\partial \boldsymbol{h}_t} = \frac{\partial \mathcal{L}}{\partial \boldsymbol{h}_T} \frac{\partial \boldsymbol{h}_T}{\partial \boldsymbol{h}_t} = \frac{\partial \mathcal{L}}{\partial \boldsymbol{h}_T} \exp \Big\{ - \int_T^t \frac{\partial f}{\partial \boldsymbol{h}} (\boldsymbol{h}(s), s, \theta) ds \Big\}.
\tag{17}
$$

- For GHBNODE[4], from (13) we can derive

$$
\begin{bmatrix} \frac{\partial \mathcal{L}}{\partial \boldsymbol{h}_t} & \frac{\partial \mathcal{L}}{\partial \boldsymbol{m}_t} \end{bmatrix} = \begin{bmatrix} \frac{\partial \mathcal{L}}{\partial \boldsymbol{h}_T} & \frac{\partial \mathcal{L}}{\partial \boldsymbol{m}_T} \end{bmatrix} \begin{bmatrix} \frac{\partial \boldsymbol{h}_T}{\partial \boldsymbol{h}_t} & \frac{\partial \boldsymbol{h}_T}{\partial \boldsymbol{m}_t} \\ \frac{\partial \boldsymbol{m}_T}{\partial \boldsymbol{h}_t} & \frac{\partial \boldsymbol{m}_T}{\partial \boldsymbol{m}_t} \end{bmatrix} = \begin{bmatrix} \frac{\partial \mathcal{L}}{\partial \boldsymbol{h}_T} & \frac{\partial \mathcal{L}}{\partial \boldsymbol{m}_T} \end{bmatrix} \exp \Big\{ - \underbrace{\int_T^t \begin{bmatrix} (\frac{\partial f}{\partial \boldsymbol{h}} - \xi \boldsymbol{I}) & \frac{\partial \sigma}{\partial \boldsymbol{m}} \\ \boldsymbol{0} & -\gamma \boldsymbol{I} \end{bmatrix} ds}_{:=\boldsymbol{M}} \Big\}.
\tag{18}
$$

Note that the matrix exponential is directly related to its eigenvalues. By Schur decomposition, there exists an orthogonal matrix $\boldsymbol{Q}$ and an upper triangular matrix $\boldsymbol{U}$, where the diagonal entries of $\boldsymbol{U}$ are eigenvalues of $\boldsymbol{Q}$ ordered by their real parts, such that

$$
-\boldsymbol{M} = \boldsymbol{Q}\boldsymbol{U}\boldsymbol{Q}^\top \Longrightarrow \exp\{-\boldsymbol{M}\} = \boldsymbol{Q}\exp\{\boldsymbol{U}\}\boldsymbol{Q}^\top.
\tag{19}
$$

Let $\boldsymbol{v}^\top := \begin{bmatrix} \frac{\partial \mathcal{L}}{\partial \boldsymbol{h}_T} & \frac{\partial \mathcal{L}}{\partial \boldsymbol{m}_T} \end{bmatrix} \boldsymbol{Q}$, then (18) can be rewritten as

$$
\begin{bmatrix} \frac{\partial \mathcal{L}}{\partial \boldsymbol{h}_t} & \frac{\partial \mathcal{L}}{\partial \boldsymbol{m}_t} \end{bmatrix} = \begin{bmatrix} \frac{\partial \mathcal{L}}{\partial \boldsymbol{h}_T} & \frac{\partial \mathcal{L}}{\partial \boldsymbol{m}_T} \end{bmatrix} \exp\{-\boldsymbol{M}\} = \begin{bmatrix} \frac{\partial \mathcal{L}}{\partial \boldsymbol{h}_T} & \frac{\partial \mathcal{L}}{\partial \boldsymbol{m}_T} \end{bmatrix} \boldsymbol{Q}\exp\{\boldsymbol{U}\}\boldsymbol{Q}^\top = \boldsymbol{v}^\top \exp\{\boldsymbol{U}\}\boldsymbol{Q}^\top.
\tag{20}
$$

By taking the $\ell_2$ norm in (20) and dividing both sides by $\big\| \begin{bmatrix} \frac{\partial \mathcal{L}}{\partial \boldsymbol{h}_T} & \frac{\partial \mathcal{L}}{\partial \boldsymbol{m}_T} \end{bmatrix} \big\|_2$, we arrive at

$$
\frac{\big\| \begin{bmatrix} \frac{\partial \mathcal{L}}{\partial \boldsymbol{h}_t} & \frac{\partial \mathcal{L}}{\partial \boldsymbol{m}_t} \end{bmatrix} \big\|_2}{\big\| \begin{bmatrix} \frac{\partial \mathcal{L}}{\partial \boldsymbol{h}_T} & \frac{\partial \mathcal{L}}{\partial \boldsymbol{m}_T} \end{bmatrix} \big\|_2} = \frac{\|\boldsymbol{v}^\top \exp\{\boldsymbol{U}\}\boldsymbol{Q}^\top\|_2}{\|\boldsymbol{v}^\top \boldsymbol{Q}^\top\|_2} = \frac{\|\boldsymbol{v}^\top \exp\{\boldsymbol{U}\}\|_2}{\|\boldsymbol{v}\|_2} = \|\boldsymbol{e}^\top \exp\{\boldsymbol{U}\}\|_2,
\tag{21}
$$

i.e., $\big\| \begin{bmatrix} \frac{\partial \mathcal{L}}{\partial \boldsymbol{h}_t} & \frac{\partial \mathcal{L}}{\partial \boldsymbol{m}_t} \end{bmatrix} \big\|_2 = \|\boldsymbol{e}^\top \exp\{\boldsymbol{U}\}\|_2 \big\| \begin{bmatrix} \frac{\partial \mathcal{L}}{\partial \boldsymbol{h}_T} & \frac{\partial \mathcal{L}}{\partial \boldsymbol{m}_T} \end{bmatrix} \big\|_2$ where $\boldsymbol{e} = \boldsymbol{v}/\|\boldsymbol{v}\|_2$.

**Proposition 4.** *The eigenvalues of $-\boldsymbol{M}$ can be paired so that the sum of each pair equals $(t - T)\gamma$.*

For a given constant $a > 0$, we can group the upper triangular matrix $\exp\{\boldsymbol{U}\}$ as follows

$$
\exp\{\boldsymbol{U}\} := \begin{bmatrix} \exp\{\boldsymbol{U}_L\} & \boldsymbol{P} \\ \boldsymbol{0} & \exp\{\boldsymbol{U}_V\} \end{bmatrix},
\tag{22}
$$

where the diagonal of $\boldsymbol{U}_L$ ($\boldsymbol{U}_V$) contains eigenvalues of $-\boldsymbol{M}$ that are no less (greater) than $(t - T)a$. Then, we have $\|\boldsymbol{e}^\top \exp\{\boldsymbol{U}\}\|_2 \geq \|\boldsymbol{e}_L^\top \exp\{\boldsymbol{U}_L\}\|_2$ where the vector $\boldsymbol{e}_L$ denotes the first $m$ columns of $\boldsymbol{e}$ with $m$ be the number of columns of $\boldsymbol{U}_L$. By choosing $0 \leq \gamma \leq 2a$, for every pair of eigenvalues of $-\boldsymbol{M}$ there is at least one eigenvalue whose real part is no less than $(t - T)a$. Therefore, $\exp\{\boldsymbol{U}_L\}$ decays at a rate at most $(t - T)a$, and the dimension of $\boldsymbol{U}_L$ is at least $N \times N$. We avoid exploding gradients by clipping the $\ell_2$ norm of the adjoint states similar to that used for training RNNs.

In contrast, all eigenvalues of the matrix $\int_T^t \partial f / \partial \boldsymbol{h} ds$ in (17) for NODE can be very positive or negative, resulting in exploding or vanishing gradients. As an illustration, we consider the benchmark Walker2D kinematic simulation task that requires learning long-term dependencies effectively [26, 3]. We train ODE-RNN [42] and (G)HBNODE-RNN on this benchmark dataset, and the detailed experimental settings are provided in Sec. 5.4. Figure 4 plots $\|\partial \mathcal{L}/\partial \boldsymbol{h}_t\|_2$ for ODE-RNN and $\|[\partial \mathcal{L}/\partial \boldsymbol{h}_t \ \partial \mathcal{L}/\partial \boldsymbol{m}_t]\|_2$ for (G)HBNODE-RNN, showing that the adjoint state of ODE-RNN vanishes quickly, while that of (G)HBNODE-RNN does not vanish even when the gap between $T$ and $t$ is very large.

---

[4]HBNODE can be seen as a special GHBNODE with $\xi = 0$ and $\sigma$ be the identity map.

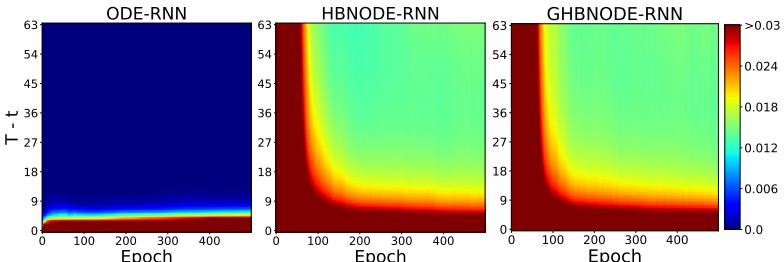

Figure 3: Plot of the the $\ell_2$-norm of the adjoint states for ODE-RNN and (G)HBNODE-RNN back-propagated from the last time stamp. The adjoint state of ODE-RNN vanishes quickly when the gap between the final time $T$ and intermediate time $t$ becomes larger, while the adjoint states of (G)HBNODE-RNN decays much more slowly. This implies that (G)HBNODE-RNN is more effective in learning long-term dependency than ODE-RNN.

## 5 Experimental Results

In this section, we compare the performance of the proposed HBNODE and GHBNODE with existing ODE-based models, including NODE [4], ANODE [10], and SONODE [35] on the benchmark point cloud separation, image classification, learning dynamical systems, and kinematic simulation. For all the experiments, we use Adam [25] as the benchmark optimization solver (the learning rate and batch size for each experiment are listed in Table 1) and Dormand–Prince-45 as the numerical ODE solver. For HBNODE and GHBNODE, we set $\gamma = \mathrm{sigmoid}(\theta)$, where $\theta$ is a trainable weight initialized as $\theta = -3$. The network architecture used to parameterize $f(\boldsymbol{h}(t), t, \theta)$ for each experiment below are described in Appendix E. All experiments are conducted on a server with 2 NVIDIA Titan Xp GPUs.

Table 1: The batch size and learning rate for different datasets.

| Dataset | Point Cloud | MNIST | CIFAR10 | Plane Vibration | Walker2D |
|---|---|---|---|---|---|
| Batch Size | 50 | 64 | 64 | 64 | 256 |
| Learning Rate | 0.01 | 0.001 | 0.001 | 0.0001 | 0.003 |

Figure 4: Comparison between NODE, ANODE, SONODE, HBNODE, and GHBNODE for two-dimensional point cloud separation. HBNODE and GHBNODE converge better and require less NFEs in both forward and backward propagation than the other benchmark models.

### 5.1 Point cloud separation

In this subsection, we consider the two-dimensional point cloud separation benchmark. A total of 120 points are sampled, in which 40 points are drawn uniformly from the circle $\|\boldsymbol{r}\| < 0.5$, and 80 points are drawn uniformly from the annulus $0.85 < \|\boldsymbol{r}\| < 1.0$. This experiment aims to learn effective features to classify these two point clouds. Following [10], we use a three-layer neural network to parameterize the right-hand side of each ODE-based model, integrate the ODE-based model from $t_0 = 0$ to $T = 1$, and pass the integration results to a dense layer to generate the classification results. We set the size of hidden layers so that the models have similar sizes, and the number of parameters of NODE, ANODE, SONODE, HBNODE, and GHBNODE are 525, 567, 528, 568,

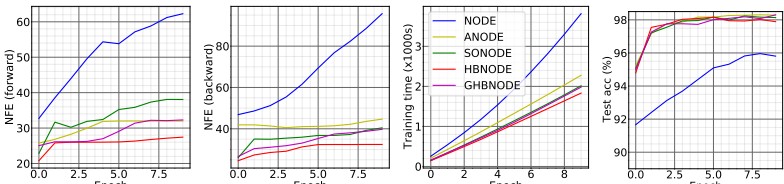

Figure 5: Contrasting NODE [4], ANODE [10], SONODE [35], HBNODE, and GHBNODE for MNIST classification in NFE, training time, and test accuracy. (Tolerance: $10^{-5}$).

and $568$, respectively. To avoid the effects of numerical error of the black-box ODE solver we set tolerance of ODE solver to be $10^{-7}$. Figure 4 plots a randomly selected evolution of the point cloud separation for each model; we also compare the forward and backward NFEs and the training loss of these models (100 independent runs). HBNODE and GHBNODE improve training as the training loss consistently goes to zero over different runs, while ANODE and SONODE often get stuck at local minima, and NODE cannot separate the point cloud since it preserves the topology [10].

## 5.2 Image classification

We compare the performance of HBNODE and GHBNODE with the existing ODE-based models on MNIST and CIFAR10 classification tasks using the same setting as in [10]. We parameterize $f(\boldsymbol{h}(t), t, \theta)$ using a 3-layer convolutional network for each ODE-based model, and the total number of parameters for each model is listed in Table 2. For a given input image of the size $c \times h \times w$, we first augment the number of channel from $c$ to $c + p$ with the augmentation dimension $p$ dependent on each method[5]. Moreover, for SONODE, HBNODE and GHBNODE, we further include velocity or momentum with the same shape as the augmented state.

Table 2: The number of parameters for each models for image classification.

| Model | NODE | ANODE | SONODE | HBNODE | GHBNODE |
|---|---|---|---|---|---|
| #Params (MNIST) | 85,315 | 85,462 | 86,179 | 85,931 | 85,235 |
| #Params (CIFAR10) | 173,611 | 172,452 | 171,635 | 172,916 | 172,916 |

**NFEs.** As shown in Figs. 1 and 5, the NFEs grow rapidly with training of the NODE, resulting in an increasingly complex model with reduced performance and the possibility of blow up. Input augmentation has been verified to effectively reduce the NFEs, as both ANODE and SONODE require fewer forward NFEs than NODE for the MNIST and CIFAR10 classification. However, input augmentation is less effective in controlling their backward NFEs. HBNODE and GHBNODE require much fewer NFEs than the existing benchmarks, especially for backward NFEs. In practice, reducing NFEs implies reducing both training and inference time, as shown in Figs. 1 and 5.

**Accuracy.** We also compare the accuracy of different ODE-based models for MNIST and CIFAR10 classification. As shown in Figs. 1 and 5, HBNODE and GHBNODE have slightly better classification accuracy than the other three models; this resonates with the fact that less NFEs lead to simpler models which generalize better [10, 35].

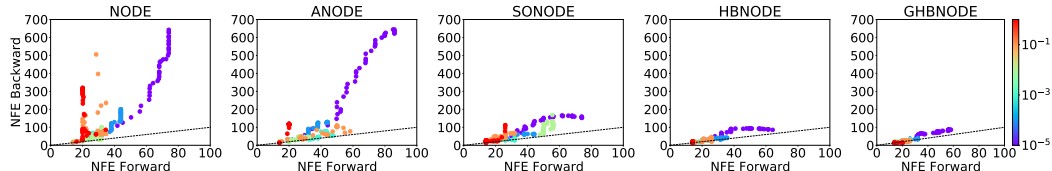

Figure 6: NFE vs. tolerance (shown in the colorbar) for training ODE-based models for CIFAR10 classification. Both forward and backward NFEs of HBNODE and GHBNODE grow much more slowly than that of NODE, ANODE, and SONODE; especially the backward NFEs. As the tolerance decreases, the advantage of HBNODE and GHBNODE in reducing NFEs becomes more significant.

**NFEs vs. tolerance.** We further study the NFEs for different ODE-based models under different tolerances of the ODE solver using the same approach as in [4]. Figure 6 depicts the forward and backward NFEs for different models under different tolerances. We see that (i) both forward

---

[5]We set $p = 0, 5, 4, 4, 5/0, 10, 9, 9, 9$ on MNIST/CIFAR10 for NODE, ANODE, SONODE, HBNODE, and GHBNODE, respectively.

and backward NFEs grow quickly when tolerance is decreased, and HBNODE and GHBNODE require much fewer NFEs than other models; (ii) under different tolerances, the backward NFEs of NODE, ANODE, and SONODE are much larger than the forward NFEs, and the difference becomes larger when the tolerance decreases. In contrast, the forward and backward NFEs of HBNODE and GHBNODE scale almost linearly with each other. This reflects that the advantage in NFEs of (G)HBNODE over the benchmarks become more significant when a smaller tolerance is used.

## 5.3 Learning dynamical systems from irregularly-sampled time series

In this subsection, we learn dynamical systems from experimental measurements. In particular, we use the ODE-RNN framework [4, 42], with the recognition model being set to different ODE-based models, to study the vibration of an airplane dataset [33]. The dataset was acquired, from time $0$ to $73627$, by attaching a shaker underneath the right wing to provide input signals, and 5 attributes are recorded per time stamp; these attributes include voltage of input signal, force applied to aircraft, and acceleration at 3 different spots of the airplane. We randomly take out $10\%$ of the data to make the time series irregularly-sampled. We use the first $50\%$ of data as our train set, the next $25\%$ as validation set, and the rest as test set. We divide each set into non-overlapping segments of consecutive 65 time stamps of the irregularly-sampled time series, with each input instance consisting of 64 time stamps of the irregularly-sampled time series, and we aim to forecast 8 consecutive time stamps starting from the last time stamp of the segment. The input is fed through the the hybrid methods in a recurrent fashion; by changing the time duration of the last step of the ODE integration, we can forecast the output in the different time stamps. The output of the hybrid method is passed to a single dense layer to generate the output time series. In our experiments, we compare different ODE-based models hybrid with RNNs. The ODE of each model is parametrized by a 3-layer network whereas the RNN is parametrized by a simple dense network; the total number of parameters for ODE-RNN, ANODE-RNN, SONODE-RNN, HBNODE-RNN, and GHBNODE-RNN with 16, 22, 14, 15, 15 augmented dimensions are 15,986, 16,730, 16,649, 16,127, and 16,127, respectively. To avoid potential error due to the ODE solver, we use a tolerance of $10^{-7}$.

In training those hybrid models, we regularize the models by penalizing the L2 distance between the RNN output and the values of the next time stamp. Due to the second-order natural of the underlying dynamics [35], ODE-RNN and ANODE-RNN learn the dynamics very poorly with much larger training and test losses than the other models even they take smaller NFEs. HBNODE-RNN and GHBNODE-RNN give better prediction than SONODE-RNN using less backward NFEs.

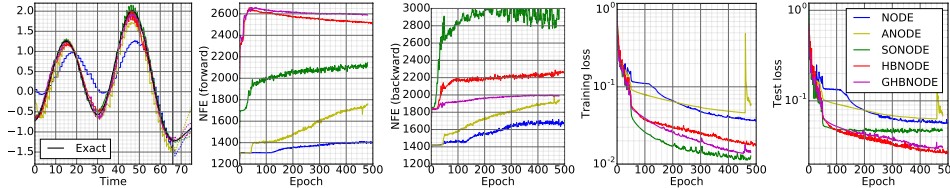

Figure 7: Contrasting ODE-RNN, ANODE-RNN, SONODE-RNN, HBNODE-RNN, and GHBNODE-RNN for learning a vibrational dynamical system. Left most: The learned curves of each model vs. the ground truth (Time: <66 for training, 66-75 for testing).

## 5.4 Walker2D kinematic simulation

In this subsection, we evaluate the performance of HBNODE-RNN and GHBNODE-RNN on the Walker2D kinematic simulation task, which requires learning long-term dependency effectively [26]. The dataset [3] consists of a dynamical system from kinematic simulation of a person walking from a pre-trained policy, aiming to learn the kinematic simulation of the MuJoCo physics engine [48]. The dataset is irregularly-sampled with $10\%$ of the data removed from the simulation. Each input consists of 64 time stamps fed though the the hybrid methods in a recurrent fashion, and the output is passed to a single dense layer to generate the output time series. The goal is to provide an auto-regressive forecast so that the output time series is as close as the input sequence shifted one time stamp to the right. We compare ODE-RNN (with 7 augmentation), ANODE-RNN (with 7 ANODE style augmentation), HBNODE-RNN (with 7 augmentation), and GHBNODE-RNN (with 7 augmentation) [6] The RNN is parametrized by a 3-layer network whereas the ODE is parametrized by a simple dense

---

[6]Here, we do not compare with SONODE-RNN since SONODE has some initialization problem on this dataset; the ODE solver encounters failure due to exponential growth over time.

network. The number of parameters of the above four models are 8,729, 8,815, 8,899, and 8,899, respectively. In Fig. 8, we compare the performance of the above four models on the Walker2D benchmark; HBNODE-RNN and GHBNODE-RNN not only require significantly less NFEs in both training (forward and backward) and in testing than ODE-RNN and ANODE-RNN, but also have much smaller training and test losses.

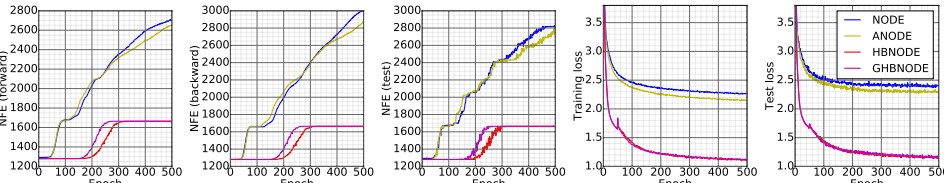

Figure 8: Contrasting ODE-RNN, ANODE-RNN, SONODE-RNN, HBNODE-RNN, and GHBNODE-RNN for the Walker-2D kinematic simulation.

## 6   Related Work

**Reducing NFEs in training NODEs.**    Several techniques have been developed to reduce the NFEs for the forward solvers in NODEs, including weight decay [15], input augmentation [10], regularizing solvers and learning dynamics [12, 23, 13, 36], high-order ODE [35], data control [29], and depth-variance [29]. HBNODEs can reduce both forward and backward NFEs at the same time.

**Second-order ODE accelerated dynamics.**    It has been noticed in both optimization and sampling communities that second-order ODEs with an appropriate damping term, e.g., the classical momentum and Nesterov's acceleration in discrete regime, can significantly accelerate the first-order gradient dynamics (gradient descent), e.g., [39, 32, 5, 45, 53]. Also, these second-order ODEs have been discretized via some interesting numerical schemes to design fast optimization schemes, e.g., [44].

**Learning long-term dependencies.**    Learning long-term dependency is one of the most important goals for learning from sequential data. Most of the existing works focus on mitigating exploding or vanishing gradient issues in training RNNs, e.g., [1, 54, 22, 51, 30, 18, 47]. Attention-based models are proposed for learning on sequential data concurrently with the effective accommodation of learning long-term dependency [50, 8]. Recently, NODEs have been integrated with long-short term memory model [19] to learn long-term dependency for irregularly-sampled time series [26]. HBNODEs directly enhance learning long-term dependency from sequential data.

**Momentum in neural network design.**    As a line of orthogonal work, the momentum has also been studied in designing neural network architecture, e.g., [31, 47, 27, 43], which can also help accelerate training and learn long-term dependencies. These techniques can be considered as changing the neural network $f$ in (1). We leave the synergistic integration of adding momentum to $f$ with our work on changing the left-hand side of (1) as a future work.

## 7   Concluding Remarks

We proposed HBNODEs to reduce the NFEs in solving both forward and backward ODEs, which also improve generalization performance over the existing benchmark models. Moreover, HBNODEs alleviate vanishing gradients in training NODEs, making HBNODEs able to learn long-term dependency effectively from sequential data. In the optimization community, Nesterov acceleration [32] is also a famous algorithm for accelerating gradient descent, that achieves an optimal convergence rate for general convex optimization problems. The ODE counterpart of the Nesterov's acceleration corresponds to (9) with $\gamma$ being replaced by a time-dependent damping parameter, e.g., $t/3$ [45] or with restart [52]. The adjoint equation of the Nesterov's ODE [45] is no longer a Nesterov's ODE. We notice that directly using Nesterov's ODE cannot improve the performance of the vanilla neural ODE. How to integrate Nesterov's ODE with neural ODE is an interesting future direction. Another interesting direction is connecting HBNODE with symplectic ODE-net [57] through an appropriate change of variables.

## 8   Acknowledgement

This material is based on research sponsored by NSF grants DMS-1924935 and DMS-1952339, DOE grant DE-SC0021142, and ONR grant N00014-18-1-2527 and the MURI grant N00014-20-1-2787.

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
