# OpenReview forum: "Heavy Ball Neural Ordinary Differential Equations"
_NeurIPS.cc/2021/Conference — NeurIPS 2021 Poster_

### Official Review · Reviewer_UXkx · 2021-07-13

**Rating:** 6
**Confidence:** 3

**Summary:**

The authors propose a new neural ordinary differential equation (NODE) model that is inspired by a second order ODE $\ddot{x} + a \dot{x} + b \nabla F(x)=0$. The discretization of this ODE leads to a two step procedure known as heavy ball method. Using this idea to model a NODE has the advantage that the corresponding adjoint equation takes a form that reduces the number of function evaluations during the forward and backward pass. Further, the authors show that the spectrum of the model is well-structured which in turn mitigates the vanishing gradient problem. In addition, they formulate a model that integrates skip connections and gating mechanisms. The empirical results demonstrate the favorable performance of the proposed models as compared to other NODEs.

**Limitations And Societal Impact:**

A detailed discussion of the limitation of the proposed models is missing (there is no Section 4.1), also there is no discussion of the potential negative societal impact. I am not qualified to judge whether there is any potential negative societal impact of this work, but since this is fundamental research, I assume that there is non foreseeable negative impact.

**Main Review:**

The originality of this work is limited, since the idea of second order neural ODEs is not new. Also, I do not agree that the work of [45] and others is orthogonal to your work (maybe you can elaborate on this). Further, you miss to discuss the work by [*1] who propose a heavy ball inspired neural network architecture. This said, it would be fair to devote more room for discussing related work within the introduction and related work section, instead of just briefly mention some of the important related works somewhere at the end of the paper.

The main contribution of this paper is the analysis of the corresponding adjoint equation for this model, and to my best knowledge these results appear to be novel. The advantage of the adjoint equation is plausible and backed up by empirical results, which show a considerable advantage as compared to other neural ODEs. Hence, I consider the results as significant and relevant for being published.

The generalized heavy ball neural ODE looks a bit heuristic and the advantage is not exactly clear to me. The experiments do not show necessarily an advantage in terms of accuracy. Based on Figure 1, I would favor the simpler heavy ball model. Does the results in Sec. 4 apply to the generalized model?

Overall, the paper appears to be very dense and thus a bit cumbersome to read. I suggest, that you move some of the details, e.g., Table 1 and 2 into the appendix. Also you probably can omit section 1.2. Personally, I would also move the discussion of the generalized heavy ball neural ODE into the appendix. In turn, this would give room to expend the introduction, better discuss related work, and provide more room for discussions and explanations.


Other comments:

* The statement that the exploding gradients issue can be effectively resolved via gradient clipping is a bit misleading. While it is true that the issue of exploding gradients can be resolved via clipping, this comes with the price of restricted expressivity. Hence, there is a lot of current research effort focusing on architectures that avoid vanishing and exploding gradients by design [*2,*3,*4]. Since, [*3,*4] are CT-RNNs, they can also handle irregular sampled time series data. I don't think that it is necessary to compare to these models, but I would suggest to slightly rephrase the discussion at beginning of Section 4.

* I am slightly surprised about the results shown in Figure 5. The augmented ODE should be able to perfectly separate the two point clouds. Are the plots here from an early stage of training or is there another reason for the poor performance of the NODE, here?

* The considered ODE models do not achieve state-of-the-art performance on CIFAR-10. For instance, ANODE-v2 [*5] and Statefull-ODEs [*6] achieve about 90% test accuracy on CIFAR-10, while having nearly the same parameter count as the models considered in the experiments. Is there any reason for why you consider only a very simple network architecture?


[*1] Li, Huan, et al. "Optimization algorithm inspired deep neural network structure design." Asian Conference on Machine Learning. PMLR, 2018.

[*2] Lezcano-Casado, Mario, and David Martınez-Rubio. "Cheap orthogonal constraints in neural networks: A simple parametrization of the orthogonal and unitary group." ICLR (2019).

[*3] Rusch, T. Konstantin, and Siddhartha Mishra. "Coupled Oscillatory Recurrent Neural Network (coRNN): An accurate and (gradient) stable architecture for learning long time dependencies." ICLR (2021).

[*4] Erichson, N. Benjamin, et al. "Lipschitz Recurrent Neural Networks." ICLR (2021).

[*5] Zhang, Tianjun, et al. "Anodev2: A coupled neural ode evolution framework." NeurIPS (2019).

[*6] Queiruga, Alejandro, et al. "Compressing Deep ODE-Nets using Basis Function Expansions." arXiv preprint arXiv:2106.10820 (2021).

**Time Spent Reviewing:**

4

---

> ### Author Response · Authors · 2021-08-09
> **Response**
>
> Thanks for your thoughtful review and the encouragement of the novelty of the analysis of adjoint equations. Below we address your concerns.
>
> -----
>
> **Q1. The originality of this work is limited, since the idea of second order neural ODEs is not new. Also, I do not agree that the work of [45] and others is orthogonal to your work (maybe you can elaborate on this). Further, you miss to discuss the work by [1] who propose a heavy ball inspired neural network architecture. This said, it would be fair to devote more room for discussing related work within the introduction and related work section, instead of just briefly mention some of the important related works somewhere at the end of the paper.**
>
> Reply: Second order neural ODEs have been studied, while our paper first studied the theoretical and empirical benefits of the damping term in second order neural ODEs, including reducing NFEs and helping to learn long-term dependencies.
>
> In the paper [45], the authors designed new ResNet-style architecture by using momentum, and the resulting architecture can be used as the right-hand side $f$ of neural ODEs. In this sense, we believe [45] and our (G)HBNODEs can be used jointly, and perhaps the advantages of both methods can be inherited and worth exploring. In this sense, we consider [45] and our method orthogonal.
>
> Thank you for pointing out the paper [*1] to us, it is indeed related and we have discussed it in our revised paper. We have also moved the related works section to the introduction as you suggested.
>
> -----
>
> **Q2. The generalized heavy ball neural ODE looks a bit heuristic and the advantage is not exactly clear to me. The experiments do not show necessarily an advantage in terms of accuracy. Based on Figure 1, I would favor the simpler heavy ball model. Does the results in Sec. 4 apply to the generalized model?**
>
> Reply: Results in section 4 apply to the generalized model, which are established for GHBNODEs. As mentioned in the footnote on page 5, HBNODE is a special case of GHBNODE.
>
> The generalized model is more stable during the first few training epochs for tasks involving long time series, so it is less likely to fail the ODE solver. It does not have a significant improvement on accuracy over HBNODE, though.
>
> -----
>
> **Q3. Overall, the paper appears to be very dense and thus a bit cumbersome to read. I suggest, that you move some of the details, e.g., Table 1 and 2 into the appendix. Also you probably can omit section 1.2. Personally, I would also move the discussion of the generalized heavy ball neural ODE into the appendix. In turn, this would give room to expand the introduction, better discuss related work, and provide more room for discussions and explanations.**
>
> Reply: Thank you for your suggestions, which indeed make the paper more readable.
>
> -----
>
> **Q4. The statement that the exploding gradients issue can be effectively resolved via gradient clipping is a bit misleading. While it is true that the issue of exploding gradients can be resolved via clipping, this comes with the price of restricted expressivity. Hence, there is a lot of current research effort focusing on architectures that avoid vanishing and exploding gradients by design [2,3,4]. Since, [3,4] are CT-RNNs, they can also handle irregular sampled time series data. I don't think that it is necessary to compare to these models, but I would suggest to slightly rephrase the discussion at beginning of Section 4.**
>
> Reply: Thanks for pointing out this along with related references. We have expanded our discussion on the exploding gradient and the solution of using the methods proposed in the above references.
>
> -----
>
> **Q5. I am slightly surprised about the results shown in Figure 5. The augmented ODE should be able to perfectly separate the two point clouds. Are the plots here from an early stage of training or is there another reason for the poor performance of the NODE, here?**
>
> Reply: It is true that ANODE can perfectly separate the two point clouds. We ran all five models with 100 different seeds, and we noticed that the loss of ANODE does not converge to approximately zero for a few seeds (see right panel of Figure 5), which we think the ANODE model gets stuck at some local minima with a bigger loss. The plot on the left-hand side of Figure 5 is randomly selected among these 100 runs.
>
> -----
>
> **Q6. The considered ODE models do not achieve state-of-the-art performance on CIFAR-10. For instance, ANODE-v2 [5] and Stateful-ODEs [6] achieve about 90% test accuracy on CIFAR-10, while having nearly the same parameter count as the models considered in the experiments. Is there any reason for why you consider only a very simple network architecture?**
>
> Reply: We used the network studied in the augmented neural ODE paper [10] as the benchmark. We have added more discussion on this and included the references you pointed out to us.
>
> -----
>
> We look forward to and appreciate your further feedback.

---

> > ### Comment · Reviewer_UXkx · 2021-08-19
> > **Thanks**
> >
> > Thanks for the detailed response. In light of the authors' comments and based on the other reviewers' comments, I feel that this paper should be considered for publication at NeurIPS.

---

> > > ### Author Response · Authors · 2021-08-19
> > > **Thank you!**
> > >
> > > Thanks for your further feedback, encouragement, and endorsement. We have also revised our paper as you suggested.

---

### Official Review · Reviewer_juca · 2021-07-14

**Rating:** 7
**Confidence:** 4

**Summary:**

This work proposes a reparameterization of a Neural ODE (NODE) as a Heavy-Ball Neural ODE (HBNODE). This is inspired by Heavy-Ball Momentum, which is the continuous limit of classical momentum.
The authors make the following claims:
The adjoint of an HBNODE is also an HBNODE.
HBNODE reduces forward and backward NFE. This is even more apparent at lower tolerances.
The spectrum of HBNODE is well-structured. This implies effective learning of long-range dependencies by avoiding vanishing gradients.
HBNODEs are more accurate
The authors verify these claims on image classification, complex dynamics, and sequential modelling.


**Limitations And Societal Impact:**

 The authors have adequately addressed the limitations and potential negative societal impact of their work.

**Main Review:**

Describe the strengths of the work:

This work clearly describes Heavy-Ball Momentum and the definition of HBNODE. The authors clearly state the use of augmented the state of a neural ODE, where the states are coupled, to solve the second-order heavy-ball ODE. The authors then generalize this by adding some tuneable/learnable weights for the terms. They claim this reduces the tendency for the hidden state to blow-up during training.
Backward NFEs are significantly reduced and this results in a reduction in training time too.
The authors demonstrate their approach on a wide variety of tasks.

Explain the limitations of this work:

It’s unclear why the results are so much weaker on the irregularly-sampled time-series experiment.
There is little explanation given for why the forward NFEs are larger for the experiment in Section 5.3 compared to the baseline, but smaller for the backward NFEs.
Are there reasons other than vanishing gradients that cause GHBNODES to perform better than other models? Does their parameterization simply make them more expressive for the given number of parameters?

Correctness:

The authors make an effort to provide a fair comparison to different baseline methods by matching the number of parameters.
The experiments are not repeated with different random seeds and averaged. Since some methods/baselines were sensitive to initialization, it could be helpful to see how strong these results are over several runs (e.g. varying random seed).
It’s not very clear from Figure 7 that (G)HBNODE has significant savings in NFE at loose tolerances.

Clarity:

The writing in the paper can be improved. In particular, Section 4 can be improved for clarity, and can be better connected to the rest of the text. If space is a concern, I think some of the math can be moved to the appendix.
Typos:
Line 272: fed through

Relation to prior work:

It is clear how this work differs from prior work.
A few suggestions on work related to Reducing NFEs for NODEs:

Opening the Blackbox: Accelerating Neural Differential Equations by Regularizing Internal Solver Heuristics
Avik Pal, Yingbo Ma, Viral Shah, Christopher Rackauckas

Learning Differential Equations that are Easy to Solve
Jacob Kelly, Jesse Bettencourt, Matthew James Johnson, David Duvenaud

Reproducibility:

Hyperparameters are given for the experiments. Given that examples are not averaged over multiple runs and it’s unclear exactly how the models were tuned for performance, there may be some issues with reproducing the results precisely.

Additional feedback, comments, suggestions:

In Figure 3 it is unclear if all the models are achieving the same performance at the task. It should be made clear what the performance is of each of the methods and if it is related to the norm of the hidden state. For example, do models with larger hidden states diverge faster?

Section 4 deals with the issue of vanishing gradients. There are no experiments which verify that vanishing gradients cause the inability to avoid vanishing gradients. The authors perform an analysis to justify regularizing the norm of the adjoint state as a method to mitigate vanishing gradients. The authors suggest clipping the norm of the adjoint state. They then have an experiment showing the GHBNODE models don’t have vanishing gradients. I think the experiment could be improved by explicitly measuring some of the quantities mentioned in the analysis in this section and seeing how they related to the norm of the adjoint state. Even a toy problem to make things very tractable could give useful insight.


**Time Spent Reviewing:**

2

---

> ### Author Response · Authors · 2021-08-09
> **Response**
>
> Thank you for your thoughtful review. Below we address your concerns.
>
> -----
>
> **Q1. It’s unclear why the results are so much weaker on the irregularly-sampled time-series experiment. There is little explanation given for why the forward NFEs are larger for the experiment in Section 5.3 compared to the baseline, but smaller for the backward NFEs. Are there reasons other than vanishing gradients that cause GHBNODES to perform better than other models? Does their parameterization simply make them more expressive for the given number of parameters?**
>
> Reply: On the irregularly-sampled time-series experiment in Section 5.3, the forward NFE of HBNODE and GHBNODE is larger than the benchmarks (NODE, ANODE, SONODE). However, there is a significant difference in their accuracy. We believe that HBNODE and GHBNODE have NFEs that are sufficient to represent the true dynamics. But the NODE and ANODE could not learn the true dynamics at all, and we believe this is due to the vanishing gradient issue, which prevents learning long-term dependencies; thus, they can have smaller NFEs.
>
> SONODE does approach a more accurate solution than NODE and ANODE, but it is still worse than HBNODE and GHBNODE in test accuracy; thus, SONODE has a forward NFE that is higher than that of NODE and ANODE, but is lower than that of HBNODE and GHBNODE. The fact that SONODE has a higher backward NFE than HBNODE and GHBNODE shows the advantage of (G)HBNODE in accelerating backward ODE solvers.
>
> Regarding if the parameterization of (G)HBNODE simply makes them more expressive for the given number of parameters, we have also tried 20 attempts for the baseline models with larger augmentation dimensions, which fails to learn the dynamics due to the initialization problem (using the benchmark initialization of the weights). Based on our experiments, the model with a larger augmentation dimension tends to suffer from the initialization problem more severely. Here, the initialization problem means that blow up exists in solving the parametrized ODE, and the adaptive numerical ODE solver cannot find any step size above machine precision to match the tolerance during the first few training epochs.
>
> -----
>
> **Q2. The authors make an effort to provide a fair comparison to different baseline methods by matching the number of parameters. The experiments are not repeated with different random seeds and averaged. Since some methods/baselines were sensitive to initialization, it could be helpful to see how strong these results are over several runs (e.g. varying random seed). It’s not very clear from Figure 7 that (G)HBNODE has significant savings in NFE at loose tolerances. Hyperparameters are given for the experiments. Given that examples are not averaged over multiple runs and it’s unclear exactly how the models were tuned for performance, there may be some issues with reproducing the results precisely.**
>
> Reply: We did 100 runs with different seeds for the point cloud separation task in Section 5.1. We did five independent runs for both MNIST and CIFAR benchmarks with different tolerance for image classification tasks. For the Walker2D experiments, we did three runs for the models.
>
> For the irregularly sampled time series task, explosion during initialization often happens for the baseline models, a phenomenon we discussed in Section 3, making it difficult to decide whether to take these initialization failure cases into account. Moreover, for the irregularly sampled time series task, due to the limitation of computation resources, we cannot do multiple entire runs on all models, especially when a very small tolerance is used; however, we have saved some early-truncated experimental data, which we could add to the appendix to show our reproducibility.
>
> For the loose tolerance cases, (G)HBNODE does not save many NFEs because there is a minimum NFEs required to solve the target ODE reliably. When the benchmarks do not use many NFEs beyond the minimum requirement, it is impossible to improve NFEs significantly. However, based on our tests, NODE often requires very high backward NFEs even using a loose tolerance, and (G)HBNODE can significantly reduce the backward NFEs.
>
> -----
>
> **Q3. The writing in the paper can be improved. In particular, Section 4 can be improved for clarity, and can be better connected to the rest of the text. If space is a concern, I think some of the math can be moved to the appendix. Typos: Line 272: fed through**
>
> Reply: Thank you for your valuable suggestion, and we have updated our paper accordingly.
>
> -----
>
> **Q4. It is clear how this work differs from prior work. A few suggestions on work related to Reducing NFEs for NODEs: "Opening the Blackbox: Accelerating Neural Differential Equations by Regularizing Internal Solver Heuristics Avik Pal, Yingbo Ma, Viral Shah, Christopher Rackauckas" "Learning Differential Equations that are Easy to Solve Jacob Kelly, Jesse Bettencourt, Matthew James Johnson, David Duvenaud"**
>
> Reply: We have discussed these methods in our revised manuscript.
>
> -----
>
> **Q5. In Figure 3 it is unclear if all the models are achieving the same performance at the task. It should be made clear what the performance is of each of the methods and if it is related to the norm of the hidden state. For example, do models with larger hidden states diverge faster?**
>
> Reply:
> We will add the performance comparison in training and test loss of different methods in our revision.
>
> Figure 3 is used only as an example to illustrate the initialization problem. Figure 3 shows that, for a long enough time series, the hidden state grows roughly exponentially in the L2 norm. This rapid speed of growth results in a numerically blow up phenomenon. In ODE theory, blow ups may occur in an ODE where there is no finite solution, e.g. the ODE $\frac{dy}{dt} = - \frac{y}{t-1}$ with initial condition $y(0)=1$ admits the unique solution $y(t) = \frac{1}{1-t}$ in interval $[0,1)$, but there is no finite solution at 1. With any positive tolerance, there is no positive step size that allows the adaptive step size solver to proceed through this point. In practice, blow ups may happen numerically. When the solution to the ODE grows too fast, the adaptive step size solver cannot find a step size above machine precision to proceed, resulting in a solver failure. Empirically, the hidden state will generally not diverge as swiftly in training if the neural network uses a small enough learning rate; however, the initialization can still be a challenge: if solver failure occurs during initialization, no training can be done.
>
> Yes, models with larger hidden states diverge faster based on our empirical observation.
>
> -----
>
> **Q6. Section 4 deals with the issue of vanishing gradients. There are no experiments which verify that vanishing gradients cause the inability to avoid vanishing gradients. The authors perform an analysis to justify regularizing the norm of the adjoint state as a method to mitigate vanishing gradients. The authors suggest clipping the norm of the adjoint state. They then have an experiment showing the GHBNODE models don’t have vanishing gradients. I think the experiment could be improved by explicitly measuring some of the quantities mentioned in the analysis in this section and seeing how they related to the norm of the adjoint state. Even a toy problem to make things very tractable could give useful insight.**
>
> Reply: Figure 4 shows that the vanishing gradient problem exists for the benchmark method (ODE-RNN) and not for (G)HBNODE-RNN, and Section 4 provides theoretical support about the reasons why there is no vanishing gradient problem for (G)HBNODE.
>
> Norm clipping or training loss regularization is used to resolve the exploding gradient problem, not the vanishing gradient issue.
>
> The Walker2d kinematic simulation is one of the benchmark tasks that require learning long-term dependencies, see “Mathias Lechner and Ramin Hasani, Learning Long-Term Dependencies in Irregularly-Sampled Time Series, arXiv:2006.04418”. We can further plot the eigenvalues of the matrices $M$ in our analysis in Section 4 besides Figures 4 for some smaller-scale problems as studied by Mathias Lechner and Ramin Hasani.
>
>
> -----
>
> We look forward to and appreciate your further feedback.

---

> > ### Comment · Reviewer_juca · 2021-08-25
> > **Reply**
> >
> > Thank you for the detailed response. This adequately addresses my concerns regarding the significance of the baseline comparisons, and other questions I had about the results. I am happy to recommend this paper for acceptance.

---

> > > ### Author Response · Authors · 2021-08-25
> > > **Thank you!**
> > >
> > > Thanks for your further feedback, encouragement, and endorsement. We have revised our paper according to your suggestions.

---

### Official Review · Reviewer_epEB · 2021-07-17

**Rating:** 7
**Confidence:** 3

**Summary:**

This is a very nice paper presenting a novel neural ODE with momentum. The idea behind the paper is solid, and the contribution is motivated by some known limitations of Neural ODEs. The authors not only provide us with compelling experimental results but also with a theoretical discussion and a lot of insights. I think the paper would benefit a lot the community.

**Limitations And Societal Impact:**

No concerning limitations, no foreseeable societal impact.

**Main Review:**

This paper is based on a simple idea: adding momentum often accelerates convergence. The authors apply this idea to neural ODEs and develop a solid theory around it: momentum is shown to help both theoretically (adjoint discussion) and experimentally.

I never published in neural ODEs – so I am not an expert – but I feel confident in recommending acceptance. The paper is totally in line with the literature (also experiment-wise) and is very well written and organized. I also did not spot any typo.

I have 2 questions for the authors

1) is it possible to translate this idea back to neural networks? i.e., what does a corresponding Resnet-like architecture look like?

2) Have you thought about applying this idea to Hamiltonian neural ODEs? Similar to the paper:

Symplectic ODE-Net: Learning Hamiltonian Dynamics with Control
Yaofeng Desmond Zhong, Biswadip Dey, Amit Chakraborty

**Time Spent Reviewing:**

1 h

---

> ### Author Response · Authors · 2021-08-09
> **Response**
>
> Thank you for your thoughtful review and endorsement. Below we address your concerns.
>
> -----
>
> **Q1. Is it possible to translate this idea back to neural networks? i.e., what does a corresponding Resnet-like architecture look like?**
>
> Reply: We used the idea of the ODE limit of the classical momentum to improve neural ODEs. There are two corresponding Resnet-like counterparts: 1) Designing neural networks that incorporate heavy ball momentum, which has been studied in e.g. [45] and the paper "Li, Huan, et al. Optimization algorithm inspired deep neural network structure design. Asian Conference on Machine Learning. PMLR, 2018" as Reviewer UXkx pointed out to us. 2) Designing neural networks based on different numerical discretization of the heavy ball ODE, perhaps the work [30] can be helpful in research along this direction. It might be interesting to explore both directions further. In particular, along the second direction, some properties of the heavy-ball ODE can be inherited by the discretized ODE and benefits Resnet-like architecture design.
>
> -----
>
> **Q2. Have you thought about applying this idea to Hamiltonian neural ODEs? Similar to the paper: "Symplectic ODE-Net: Learning Hamiltonian Dynamics with Control Yaofeng Desmond Zhong, Biswadip Dey, Amit Chakraborty".**
>
> Reply: Thanks for pointing out this paper to us, which is quite exciting and insightful. There is a connection between HBNODE and symplectic ODE-Net. HBNODE satisfies
> $$
> \frac{d^2h}{dt^2} = f(h, t) - \gamma \frac{dh}{dt}
> $$
> by the following change of variable
> $$
> w = e^{\gamma t/2} h,\  g(w, t) = e^{\gamma t/2} f (e^{-\gamma t/2} w, t)
> $$
> we have
> $$
> \frac{d^2w}{dt^2} = \frac{\gamma^2}{4} w + g(w, t),
> $$
> which we call the interaction form of HBNODE.
>
> The symplectic ODE-Net satisfies the following equation
> $$
> H = \frac{1}{2} p^\top M^{-1}(q) p + V(q),
> $$
> with $\frac{dq}{dt} = \frac{\partial H}{\partial p}$,  $\frac{dp}{dt} = -\frac{\partial H}{\partial q}$; we can rewrite it in terms of a second order ODE of $q$ as
> $$
> \frac{d^2 q}{dt^2} = \frac{d \frac{dH}{dp}}{dt} = \frac{dM^{-1}(q)p}{dt},
> $$
> when $M$ is independent of $q$, this equation becomes the same as the interaction form of HBNODE ($q:=w$) if
> $$
> \frac{\partial V}{\partial q} = -M( \frac{\gamma^2}{4} q + g(q, t)).
> $$
> The above equation is one of the reasons why we added a residual term in GHBNODE, which will cancel out the negative semi-definite effect of the term $-M \frac{\gamma^2}{4} q$.
>
> Symplectic ODE-Net points out a further direction in generalizing HBNODE. The $q$-dependence of $M$ might be good to be incorporated into HBNODE in our further studies. The properties of $V$ will provide more insights for the choice of neural networks $g$ and $f$. The idea of incorporating control is interesting, and the angle embeddings might provide a lot more improvement on the angle-based datasets.
>
> In most Hamiltonian systems, HBNODE should provide good performance, but probably not as good as symplectic ODE-Net, which is designated towards these problems. HBNODE might provide some directions on generalizing symplectic ODE, e.g. coupling HBNODE with symplectic ODE and use HBNODE to predict the control term might be a promising direction. Also, we can think about a direction to work on relaxing the requirements of Hamiltonian systems with the idea of HBNODE: if $-\epsilon H \leq \frac{dH}{dt} \leq 0$, we will have a control over the decay of Hamiltonian (well, it may not be called a Hamiltonian any more). We have added this insight and reference into our revised manuscript.
>
> -----
>
> We look forward to and appreciate your further feedback.

---

> > ### Comment · Reviewer_epEB · 2021-08-19
> > **Thanks!**
> >
> > Thanks, authors for the detailed explanation. I will keep my score and wish you good luck. Please make sure to incorporate the interesting comments of this rebuttal in a potential camera ready.

---

> > > ### Author Response · Authors · 2021-08-19
> > > **Thank you!**
> > >
> > > Thanks for your further feedback, encouragement, and endorsement. We have already added these insights to our revised paper.

---

### Decision · Program_Chairs · 2021-09-27

**Decision:**

Accept (Poster)

**Comment:**

The paper proposes a reparameterization of a Neural ODE (NODE) as a Heavy-Ball Neural ODE (HBNODE) in order to handle the notoriously difficult problem of long-range dependency modeling. Proposed approach is elegant and novel. The authors provided also very comprehensive empirical evaluation showcasing the effectiveness of their method and addressed *in detail* all reviewers' comments.